# Association between Fecal Short-Chain Fatty Acid Levels, Diet, and Body Mass Index in Patients with Inflammatory Bowel Disease

**DOI:** 10.3390/biology11010108

**Published:** 2022-01-10

**Authors:** Agnieszka Dąbek-Drobny, Olga Kaczmarczyk, Michał Woźniakiewicz, Paweł Paśko, Justyna Dobrowolska-Iwanek, Aneta Woźniakiewicz, Agnieszka Piątek-Guziewicz, Paweł Zagrodzki, Małgorzata Zwolińska-Wcisło

**Affiliations:** 1Unit of Clinical Dietetics, Department of Gastroenterology and Hepatology, Jagiellonian University Medical College, 30-688 Krakow, Poland; a.dabek-drobny@uj.edu.pl; 2Department of Gastroenterology and Hepatology, Jagiellonian University Medical College, 30-688 Krakow, Poland; okaczmarczyk@su.krakow.pl (O.K.); agnieszka.guziewicz@onet.eu (A.P.-G.); 3Department of Analytical Chemistry, Faculty of Chemistry, Jagiellonian University, 30-387 Krakow, Poland; michal.wozniakiewicz@uj.edu.pl (M.W.); aneta.wozniakiewicz@uj.edu.pl (A.W.); 4Department of Food Chemistry and Nutrition, Jagiellonian University Medical College, 30-688 Krakow, Poland; p.pasko@uj.edu.pl (P.P.); justyna.dobrowolska-iwanek@uj.edu.pl (J.D.-I.); pawel.zagrodzki@uj.edu.pl (P.Z.)

**Keywords:** inflammatory bowel disease, short-chain fatty acids, BMI, isovaleric acid, isobutyric acid, diet, low-fiber diet

## Abstract

**Simple Summary:**

Inflammatory bowel disease (IBD) is a chronic disorder of the gastrointestinal tract associated with gut inflammation and a disturbance in the amount of bacteria living in the human intestines. As a result, there is a reduction in the production of bacterial metabolites, especially short-chain fatty acids (SCFAs), which are formed from dietary fiber. The aim of our study was to assess the relationship between body mass index (BMI), the type of diet used, and changes in fecal SCFA levels in patients with IBD. We enrolled 61 patients with IBD and 16 patients as a control group. We asked all participants about their daily diet, using the simplified FF questionnaire, and measured the levels of SCFA in their stool samples. Our results revealed that underweight subjects had higher levels of isobutyric acid, whereas those with excessive weight had lower level of butyric, isovaleric, and propionic acids. Furthermore, we observed higher levels of valeric acid in controls than in IBD patients. However, we did not observe a relationship between diet habits and fecal SCFA levels. In conclusion, we demonstrated that BMI is associated with SCFA levels in patients with IBD.

**Abstract:**

Disturbances in the production of bacterial metabolites in the intestine have been reported in diseases associated with dysbiosis, such as inflammatory bowel diseases (IBDs) that include two conditions: Crohn disease (CD) and ulcerative colitis (UC). Short-chain fatty acids (SCFAs) are the main dietary-fiber-derived bacterial metabolites associated with the course of intestinal inflammation. In this study, we assessed the relationship between body mass index (BMI), the type of diet used, and changes in fecal SCFA levels in patients with IBD. We performed nutritional assessments using a nutritional questionnaire and determined fecal SCFA levels in 43 patients with UC, 18 patients with CD, and 16 controls. Our results revealed that subjects with a BMI > 24.99 kg/m^2^ had higher levels of isobutyric acid, whereas those with a BMI < 18.5 kg/m^2^ had lower level of butyric, isovaleric, and propionic acids. Furthermore, we observed higher levels of valeric acid in controls than in IBD patients. We did not reveal a relationship between a specific SCFA and the type of diet, but eating habits appear to be related to the observed changes in the SCFA profile depending on BMI. In conclusion, we demonstrated that BMI is associated with SCFA levels in patients with IBD.

## 1. Introduction

Inflammatory bowel disease (IBD) is an umbrella term for some incurable chronic inflammatory conditions of the gastrointestinal tract. The two most common, namely, Crohn disease (CD) and ulcerative colitis (UC), are associated with disturbance of the intestinal microbiota (dysbiosis) [1,2]. The gut microbiota plays an important role in the regulation of intestinal homeostasis, including maturation and the functioning of epithelial and immune cells [3]. Recent studies have emphasized that dysbiosis is related to the pathogenesis of IBD and may exacerbate the course of the disease [2,3]. Changes in the gut microbiota have been frequently observed in patients with IBD, with certain changes clearly linked to either CD or UC: the most consistent alteration is a reduction in Firmicutes [4]. There is a notion that there may be some reorganization of *Bacteroides* species in patients with IBD, with *Bacteroides fragilis* having a greater proportion of the bacterial mass in patients with IBD compared with controls [5]. Changes in the two dominant phyla, Firmicutes and Bacteroidetes, are connected with a growth in many members of the Proteobacteria phylum, which have been increasingly accepted to have a key role in IBD [6]. Moreover, an increasing body of research has reported changes in the production of bacterial metabolites, including short chain-fatty acids (SCFAs), bile acids, and amino acids in IBD patients [7,8,9]. SCFAs, such as butyric, propionic, acetic, and valeric acids, are carboxylic acids produced through aerobic fermentation of dietary fiber in the intestine. It is well known that SCFAs have an important function in maintaining human health, including gut homeostasis, and serving as the main fuel for the colonic epithelial cells. SCFAs display anti-inflammatory activity and are responsible for strengthening the intestinal barrier, which prevents leaky gut syndrome [10,11,12]. Due to these properties, they seem to be particularly beneficial in patients with intestinal inflammation. Among SCFAs, butyric acid has the best proven beneficial effect, whereas the importance of the other acids remains incompletely understood [13,14]. In addition to SCFAs, which are the main bacterial products, other organic acids are formed by the gut microbiota. Their role is less known. Among them, we distinguish isobutyric and isovaleric acids, collectively called branched short-chain fatty acids (BCFAs), which are produced through the fermentation of branched chain amino acids [15]. In contrast to SCFAs, increased levels of BCFAs have an unfavorable effect on gut health [16]. However, the relationship between the role of individual bacterial-derived acids and intestinal inflammation remains unclear. Moreover, the association between factors influencing the onset and course of IBD such as eating habits, lifestyle, medications, and the profile of bacterial-derived acids, is not fully understood.

The crucial factors necessary for proper SCFA production are the consumption of dietary fiber every day and microbial balance associated with the presence of SCFA-producing bacteria in the intestinal microbiota [11]. In contrast, BCFAs are formed by intestinal bacteria in the case of carbohydrate deficiency and protein breakdown [17]. Environmental factors, such as the use of antibiotics and diet modifications, including reducing the amount of fiber in the diet, together with dysbiosis, may lead to a reduction in SCFA production [18,19,20]. In previous research, particular attention has been paid to the SCFA imbalance related to the disturbance of the microbiota in IBD patients, whereas the influence of changes in eating habits and unstable body weight on the production of bacterial-derived fatty acids has not been thoroughly investigated. This seems particularly important due to the fact that during the course of IBD, patients follow a variety of diets, including low-fiber and elimination diets, such as milk-free, gluten-free, and easy-digestible diets, to relieve symptoms or for fear of their recurrence [21]. This can lead to disturbances in the production of bacterially-derived acids and, paradoxically, aggravate intestinal inflammation and exacerbate symptoms [22]. In addition, IBD patients are exposed to fluctuations in body weight during the course of the disease, which is associated with a disturbance of the intestinal microbiome [23]. Body mass index (BMI) is the best-known indicator used to assess body weight [24]. One of the serious complications of IBD is malnutrition, which is manifested by significant weight loss and lower BMI and related to disturbances of the intestinal microbiota [25]. Low BMI may be linked to more serious complications during treatment [26]. On the other hand, IBD patients are at risk of rapid weight gain caused by treatment such as glucocorticoid therapy [27]. In recent years, there has been an increase in overweight and obesity in patients with IBD [28]. It is estimated that approximately 15–40% of IBD patients are obese [25]. Some studies show that the microbiome plays a vital role in host energy homeostasis and the development of obesity and its associated metabolic disorders, although the specific mechanisms remain unclear [29]. Understanding the impact of underweight and obesity on the production of SCFAs and BCFAs will allow the introduction of more effective nutritional therapy for IBD in the future.

The aim of our study was to investigate an association between body mass index (BMI), rapid changes in body weight, and alterations in the concentration of bacteria-derived fatty acids in the stools of IBD patients. Moreover, we aimed to examine the relationship between the type of diet used (including a fiber-restricted diet), age, and the acid profile. In addition to the most commonly determined butyric, propionic, and acetic acids, we measured the levels of valeric, isovaleric, and isobutyric acids.

## 2. Materials and Methods

### 2.1. Study Population

A total of 77 participants were enrolled in the study, including 43 patients with UC, 18 patients with CD, and 16 controls. The control group included patients with functional bowel disorders, without abnormalities on lower gastrointestinal endoscopy, and who did not meet the Rome IV diagnostic criteria for irritable bowel syndrome. The age median of IBD patients was 32.0 years, and of controls, 23.5 (Table 1). The diagnosis of UC and CD was based on clinical evaluation and a combination of endoscopic, biochemical, histological, and/or radiological parameters. The assessment of disease activity was carried out according to the criteria of the Mayo Clinic in UC patients and the Crohn’s Disease Activity Index in CD patients. In CD, intestinal inflammation affected the large intestine. IBD patients included both patients in remission (14 UC, 8 CD) and with active disease (29 UC, 10 CD). During the study, IBD patients continued their current therapy, including steroids (28 patients), immunosuppressive therapy (20 patients), 5-aminosalicylic acid (56 patients), antibiotics (19 patients), and anti-tumor necrosis factor-alpha therapy (12 patients).

All participants were of Polish nationality and completed a clinical interview, a questionnaire detailing clinical and nutritional information, and provided a stool sample. The questionnaire was completed by all subjects on the day the stool sample was collected. The exclusion criteria were as follows: pregnancy, any malignancies, acute infections, alcohol and drug abuse, intake of medications that may affect SCFA production (including pre- and probiotics), partial and total parenteral nutrition, and serious somatic disorders not related with IBD.

BMI was calculated for all participants of the study by dividing a person’s weight by their height squared (kg/m^2^). It ranged from 13.8 to 37.1 kg/m^2^. For statistical purposes, participants were classified into three categories: underweight (BMI < 8.5 kg/m^2^), normal weight (BMI 18.5–24.99 kg/m^2^), and excessive weight (BMI > 24.99 kg/m^2^).

Written informed consent was obtained from all the subjects before the beginning of the study. The study protocol was approved by the local bioethical committee at Jagiellonian University (no. KBET 1072/6120/18/2018) and conducted in accordance with the Declaration of Helsinki.

### 2.2. Questionnaire

The participants of the study completed a questionnaire, which was divided into two parts. The first part concerned demographic and medical data, such as sex, age, disease duration, comorbid diseases, and medication use. All participants were asked about drugs used in the last 3 months, both those related to IBD and drugs used for a different reason than IBD, including over-the-counter medicines, vitamins, and dietary supplements. The use of probiotics and prebiotics was also assessed. In addition, all subjects were asked about their change in body weight over the last 6 months before the study. The participants were then divided into three groups: weight loss (>5% current body weight), stable body weight (changes in body weight up to 5%), and weight gain (>5% current body weight).

The second part of the questionnaire concerned dietary information. All participants were asked about their current diet used for the last 3 months before the study. The nutritional questionnaire was an adaptation of the dietary assessment tool, the Food Frequency Questionnaire (FFQ), which originally qualitatively estimated the frequency of consumption of 62 product groups. For the purposes of this study, we estimated the frequency of consumption of selected product groups, including fresh vegetables and fruits, dried fruits, whole-grain products, legumes, alcohol, and dairy products. Based on the responses to the questionnaire, four groups of subjects were distinguished for each category, depending on the frequency of intake of products. The subjects were divided into the following consumption groups: several times a day, several times a week, several times a month, and no consumption, for each of the selected product groups. Moreover, the frequency of fiber consumption was assessed according to the responses of the participants. The subjects were assigned to the three groups: normal-, low-, and high-fiber consumption, depending on the type of diet used.

### 2.3. Identification and Determination of Fecal Organic Acids

To identify and determine the concentration of fecal organic acids, we used the method described in a previous study [30]. We collected stool samples in sterilized plastic cups with lids and stored them at −80 °C for further analysis. The sample preparation and extraction processes were carried out at the Department of Food Chemistry and Nutrition, Faculty of Pharmacy, Jagiellonian University Medical College, Krakow, Poland. Preparation process included drying, milling, and subsequent extraction process. We optimized the extraction conditions by selecting 4 min for the shaking time, 40 min for the ultrasonic exposure of the sample, and three repetitions of the extractions. Then, all extracts were centrifuged and stored at −20 °C until further analysis.

The short-chain fatty acid pilot investigations in stool extracts were made using an iso-tachophoresis system (Electrophoretic Analyser EA 202 M, Villa Labeco, Spisska Nova Ves, Slovakia) with a conductivity detector.

Capillary electrophoresis with spectrophotometric detection (CE-UV) was used for the determination of the following acids in stool samples: acetic, propionic, butyric, isobutyric, valeric, and isovaleric [22]. Electrophoretic measurements were carried out with the PA 800 CE apparatus plus the Pharmaceutical Analysis System (Beckman-Coulter, Brea, CA, USA), equipped with an ultraviolet spectrophotometric detector, at the Laboratory for Forensic Chemistry, Faculty of Chemistry, Jagiellonian University. The separation of the analyzed compounds was carried out in a fused silica capillary (75 µm ID, 60 cm total length/50 cm effective length) at 25 °C with −30 kV applied. The injection was made hydrodynamically by applying a pressure of 3.45 kPa for 8 s. The indirect spectrophotometric detection was performed at 230 nm. The separation buffer was composed of 1% of methyl-β-cyclodextrin in a commercially available buffer (Anion Kit 5, Analis, Namur, Belgium). The Anion Kit 5 was used for the qualitative and quantitative analysis of anions of inorganic and organic acids (e.g., chloride, azide, formate, succinate, acetate, propionate, butyrate, valerate, caproate). The modification of the commercially separation buffer with cyclodextrin additionally allowed the separation of butyric acid anions from isobutyrate and valerate from isovaleric acid anions. The limit of quantification (LOQ) was 26 ug/g and was taken as the lowest concentration of standard solution of calibration curves measured for a given analyte (7.8 ug/mL), taking into account the mass of the sample, 0.3 g. The limit of determination (LOD) was calculated as the value three times lower than the LOQ for the given acid. The precision was calculated as the RSD of repeated measurements (n = 9) and did not exceed 15%.

### 2.4. Statistical Analysis

All parameters were gathered in a contingency table, together with FA fecal concentrations in the individuals. The latter parameters were then transformed into the simplest ordinal scale (two categories) according to their values being either under or above the respective medians. All these parameters created a multidimensional space of original scores. For these scores, a correspondence analysis model (CA) was constructed under the condition that its first two dimensions should explain at least 50% of the total inertia in the original set of parameters. Thus, the parameters with the lowest quality of representation were subsequently discarded. Those parameters with large absolute values of their coordinates (>0.3) in the CA model were assumed to be associated with one another. To express the strength of bivariate associations, for the pairs of associated parameters, the algebraic products of their corresponding coordinates and the cosine of the corresponding angle were calculated (these coefficients are called the association weights). The “corresponding angle” was defined as the angle determined by two lines connecting the origin with coordinates of both parameters on the CA coordinates plot.

Statistical analysis was conducted using the package STATISTICA v. 13.3 (TIBCO Software Inc., Palo Alto, CA, USA). The program provided by MP System Co. (Chrzanów, Poland) was used to calculate correlation weights for parameter pairs showing correlations in the CA model. The software available at http://statpages.org/ctab2x2.html, accessed on 1 December 2021, was used to calculate odds ratios. A probability level of *p* less than 0.05 was considered to be significant.

## 3. Results

### 3.1. Type of Diet in the Study Groups

The types of diet followed by patients with UC, CD, and controls are presented in Figure 1. The most common diet used in UC patients was a low-fiber diet (Figure 1A), whereas in CD patients, it was a regular diet and then a low-fiber diet (Figure 1B). There were no significant differences in the proportion of patients with UC and CD using standard and low-residue diets. Similarly, for these two groups, there was no difference between the proportion of patients using a standard diet and all other patients, nor between consumers of a low-residue diet and those of all other diets. The UC group differed from controls with respect to the consumption of a standard diet and other diets (odds ratio (OR) = 0.269 [95% confidence interval (CI): 0.079–0.917, *p* = 0.042]).

The results regarding the consumption of diets with different fiber contents in various groups of patients are presented in Figure 2. In the UC group, the most common diet was a low-fiber diet, whereas in the CD group, this type of diet was the second most common after normal fiber consumption (OR not significant).

The consumption of diets with different fiber contents in the groups of underweight, normal-weight, and excessive-weight subjects are presented in Figure 3. We showed that underweight subjects most often followed a low-fiber diet (Figure 3). The ORs for results obtained for the second and third BMI category groups were not significant.

### 3.2. Correspondence Analysis Model

The results of correspondence analysis showed associations between different BMI groups and the level of selected SCFAs and BCFAs (Table 2, Figure 4). Excess body weight (BMI > 25 kg/m^2^) was associated with a higher level of isobutyric acid, whereas low body weight (BMI < 18.5 kg/m^2^) was associated with a lower level of butyric, isovaleric, and propionic acids. However, we did not observe a relationship between the concentration of individual fatty acids with age and diet. We found that the age of subjects was associated with dietary pattern (Table 3). We also showed that younger subjects (<30 years of age) consumed legumes and whole grains more often, and they were more likely to follow a standard diet. In addition, younger subjects were less likely to be underweight, as there was an inverse relationship between young age and underweight status. Furthermore, a high-fiber diet was associated with legume consumption several times a week.

The results of the correspondence analysis showed that underweight patients were more often treated with steroids and antibiotics (Table 4). In addition, being treated with steroids was associated with avoidance of whole-grain products and legumes in the diet (Table 4).

### 3.3. Association between the Concentration of Fatty Acids, Consumption of Some Products, and Study Groups

We revealed that higher concentrations of valeric acid were more common in the control group than in IBD patients (OR = 8.883; 95% CI: 2.270–34.766, *p* = 0.001). In addition, we found that the control group had a higher incidence of legume consumption (OR = 4.181; 95% CI: 1.088–16.063, *p* = 0.041), and alcohol (OR = 14.800; 95% CI: 2.950–74.241, *p* = 0.0001).

## 4. Discussion

Our study revealed new relationships between body weight, eating habits, and SCFA and BCFA profiles, as well as IBD. Moreover, our findings revealed an association between BMI and fecal SCFA and BCFA levels. We noted that overweight and obese subjects had higher levels of isobutyric acid, whereas underweight subjects had lower levels of butyric, isovaleric, and propionic acids. In addition, we observed that a higher level of valeric acid occurred in controls than in patients with IBD. Finally, we found that the age of the subjects and the course of IBD requiring steroid therapy were associated with a specific pattern of dietary fiber consumption.

In our study, we demonstrated that underweight subjects (BMI < 18.5 kg/m^2^) had a lower level of butyric, propionic, and isovaleric acids. We showed that three quarters of the underweight subjects were on a low-fiber diet, which may explain the low level of SCFAs, including butyric and propionic acids, but not BCFAs. The accumulated literature confirms that dietary fiber promotes many beneficial bacteria and suppresses potentially harmful species, as well as stimulating SCFA production [31,32]. Furthermore, previous studies showed that with healthy gut microbiota, high fiber consumption from various sources (arabinoxylan, β-glucan, β-fructans, pectin, and cellulose) is associated with higher SCFA production [33,34]. This is in line with our findings. On the other hand, there is evidence that the restriction of fiber intake, mainly insoluble fiber, leads to an increase in BCFA production, which is the opposite of our observation [35]. It is possible that malnourished and dysbiotic IBD patients with increased protein requirements due to protein deficiency do not present increased protein fermentation and BCFA production. Hence, low isovaleric acid levels may be a marker of malnutrition in IBD patients. Presumably, the use of prebiotics and probiotics, as well as an adequate supply of protein, can jointly lead to the normalization of protein and carbohydrate fermentation, balance in the production of SCFAs and BCFAs, and, consequently, to the recovery of IBD patients. Moreover, numerous studies have emphasized that IBD patients are frequently malnourished, which may be influenced by their dietary restrictions, such as their lower fiber intake, as shown by our research [20]. All this evidence suggests that recommendations to limit dietary fiber, as proposed in the IBD dietary guidelines, often prescribed in patients with diarrhea to enhance digestive comfort, should be re-evaluated [34].

Moreover, we showed that one of the BCFAs, namely, isobutyric acid, had a significantly higher value in subjects with BMI > 24.99 kg/m^2^. One of the possible explanations for this observation may be the nutritional modifications associated with changing the diet and the elimination of selected products in this group of patients. However, only a quarter of subjects with excessive weight followed a low-fiber diet. Hence, unlike previous studies, we did not correlate high BCFA production with dietary fiber restriction [35]. In addition, despite the fact that we did not estimate protein intake in this study, none of the study participants reported being on a high-protein diet. Another possible explanation could be the imbalance in the microbial community of the gut. Besides IBD, obesity is another condition associated with dysbiotic gut microbiota [36]. Recent research has suggested that excess fat modulates both beneficial and potentially harmful microbes and, as a result, increases the ratio of Firmicutes to Bacteroides in the gut [37,38]. However, BCFAs appear to be produced by bacteria belonging to the genera *Bacteroides* and *Clostridium*, but the specific species responsible for their production are still unknown [39]. The incidence of IBD increases with overweight and obesity. In spite of popular belief, approximately 15–40% of IBD patients are obese, which may contribute to the development of IBD [28]. On the other hand, IBD may be an independent risk factor for the development of obesity due to dysbiosis and changes in intestinal microbial metabolism [40,41]. Previous research shows that changes in the immune system lead to metabolic disease [40]. Moreover, treatment of IBD, such as corticosteroid use, may also contribute to overweight [28]. On the other hand, obesity is associated with dysbiosis and changes in intestinal barrier function and leads to altered intestinal immunity [42,43], and fat tissue might promote a pro-inflammatory tendency towards IBD [40].

It seems possible that the higher concentration of isobutyric acid observed by us, with the simultaneous lack of changes in the SCFA profile, resulted from reduced BCFA-producing bacteria caused by both IBD and excessive body weight. Perhaps this could play a role as a disease marker in obese patients with IBD. To the best of our knowledge, no previous studies have investigated the relationships of SCFA and BCFA profiles in IBD patients by BMI category. A recent study on healthy individuals reported that total SCFA and propionate levels were higher in obese subjects than in lean subjects, but no difference was found for BCFA [37]. Hence, the discrepancy in our observations may be due to an IBD-related microbiome disbalance that promotes protein fermentation and isobutyric acid production. Nevertheless, elevated levels of isobutyric acid in IBD patients may be associated with the production of harmful products such as ammonia and phenols that exacerbate intestinal inflammation [44].

We also noted a higher concentration of valeric acid in the control group. Our results are in line with previous reports showing that in healthy subjects, when the microbiota is diversified, there is an increased production of longer SCFAs [45]. There is growing evidence confirming its beneficial role in intestinal inflammation, including regulation of the immune response, although the exact mechanism of its action is still under investigation [46,47]. Dietary interventions that restore healthy gut microbiota and raise valeric acid levels appear to be effective in maintaining and restoring remission in IBD patients.

In our study, we did not observe any relationship between the rapid change in body weight—either of weight gain or loss—or age and SCFA and BCFA profile. Contrary to our results, a recent study found that after weight loss interventions in obese people, the total amount of SCFAs was reduced, whereas that of BCFAs increased [44]. This discrepancy may be caused by the fact that IBD is a disease associated with microbiome changes, as well as various dietary modifications in these patients, as observed herein, that affected acid production.

Moreover, we found that IBD patients used different diets. There are dietary recommendations for the active phase of IBD, but no recommendations for the nonactive phase of this disease have been fully described to date. The European guidelines assume that during an exacerbation patients should follow an easily-digestible diet. However, the period of remission is generally defined as “healthy eating”, which could explain the discrepancy in patients’ diets [48]. In our study, 56% of patients with UC and 39% of those with CD were on an easily digestible diet, whereas none of the controls used this diet. It is even more interesting that as many as 37% of UC and 44% of CD patients were on a standard diet. With the development of knowledge about nutrition, low-fiber diets in IBD treatment have shifted towards new nutritional therapies, as can be seen among young research participants. Our results revealed that young patients more often decided to return to a normal diet, consisting of all product groups without any eliminations, which meant the consumption of fiber and legumes. This may be related to the fact that the National Center of Nutritional Education, the Polish institution setting nutritional standards for Poles, places a great emphasis on the education of young people and on modern and more easily accessible forms of information transmission. According to other investigators, IBD patients should not chronically follow a low-fiber diet [34]. The International Organization for the Study of Inflammatory Diseases recommends the consumption of fruit and vegetables by CD patients, with the exception of patients with fibrotic gastrointestinal obstruction, who should limit their intake of insoluble fiber. However, there are insufficient scientific data to recommend any specific dietary change or restriction of fruit and vegetables in patients with UC [43].

It is worth emphasizing that many IBD patients pay attention to their diet and associate the occurrence of symptoms with specific products. This is extremely important, as it was reported that an improper diet is clearly associated with the occurrence and severity of IBD [49]. In other studies, patients with IBD often used a gluten-free diet and a low fermentable oligosaccharides, disaccharides, monosaccharides, and polyols (FODMAP) diet, which was not used by our participants [50,51]. As mentioned above, such a multitude of diets clearly shows the need to clarify the guidelines for nutrition in patients with remission and exacerbation, as well as to pay attention to not excluding certain beneficial products from the diet. For example, eating small amounts of fiber in favor of refined sugars or particular starches in the diet contributes to enhanced pathogenic bacteria in the gut lumen [52]. SCFAs can improve the tolerance of more foods in the patient’s diet; therefore, according to our results, people who eat more fiber in their diet tolerate legumes more often, which may be due to the favorable composition of the intestinal microbiota due to dietary fiber intake [52].

We found that treatment with steroids is inversely related to the consumption of legumes and fiber. This dependence can be two-sided, because people who eat less fiber in their diet may have altered microbiota and lower SCFA production, which means they are exposed to frequent and severe exacerbation of IBD. On the other hand, steroid use may cause changes in the microbiota and its metabolites, which make those patients less tolerant to fiber and legume intake in their everyday diet [50,53]. In addition, steroids are used in exacerbation of the disease, so intestinal inflammation itself may cause worse tolerance of fiber-rich products. According to previous research, the microbiota has a beneficial effect on the human colon by creating a nutrient-rich environment, but on the other hand, an inappropriate diet, steroids, and lifestyle can induce changes in microbiota composition [11]. Nutritional status is crucial to healthy lifestyle and ensures the appropriate functioning of the gut microbiota, its metabolites, and the immune system.

To quantify any dependencies between the investigated parameters, including the nominal or ordinal data from the questionnaire (dichotomous and categorical parameters characterizing the patients) a statistical correspondence analysis method was implemented. The analyses of parameters’ coordinates in the coordinate system of the correspondence analysis model, generated in the reduced (two-dimensional) space determined by the first two new dimensions of this model, allowed us to reveal the structure of associations between parameters.

IBD affects a group of heterogeneous patients, characterized by different locations of the disease, various clinical courses, and thus different applied treatments. In addition, due to the lack of explicit dietary recommendations, IBD patients follow various diets. The application of the statistical correspondence analysis method allowed us to take into account all these factors influencing the production of SCFA. The associations we observe will allow us to conduct further, more detailed research in the future.

Our research has several limitations. We qualitatively estimated the frequency of consumption of several product groups, according to the modified Food Frequency Questionnaire, and we did not make an accurate assessment of the consumption of protein-rich products. On the other hand, we assessed the type of diet used by the participants, which allowed us to separate patients on a low or high protein diet, but we relied on an interview collected from patients, not on a food diary. The results of this study allowed us to identify new relationships that we intend to confirm by using a quantitative approach to dietary assessment in a large cohort of patients.

## 5. Conclusions

In conclusion, an altered SCFA profile is associated with BMI in IBD patients. Our results support the hypothesis that the regular consumption of a variety of fiber-rich foods is a factor in maintaining the proper balance of these acids. Moreover, we revealed that an abnormal BMI is associated with a disturbance in the SCFA and BCFA profile and can make the treatment of IBD more difficult. Our findings reveal the significance of valeric, isovaleric, and isobutyric acids in IBD, which until now was poorly understood. A higher concentration of valeric acid occurs in healthy people and is most likely an indicator of a healthy microbiome. According to our results, it seems that isovaleric acid may be a fecal marker of malnutrition in IBD patients, although further clinical studies are needed to investigate its role in intestinal inflammation. Finally, it seems important to update the nutritional standards for IBD patients.

## Figures and Tables

**Figure 1 biology-11-00108-f001:**
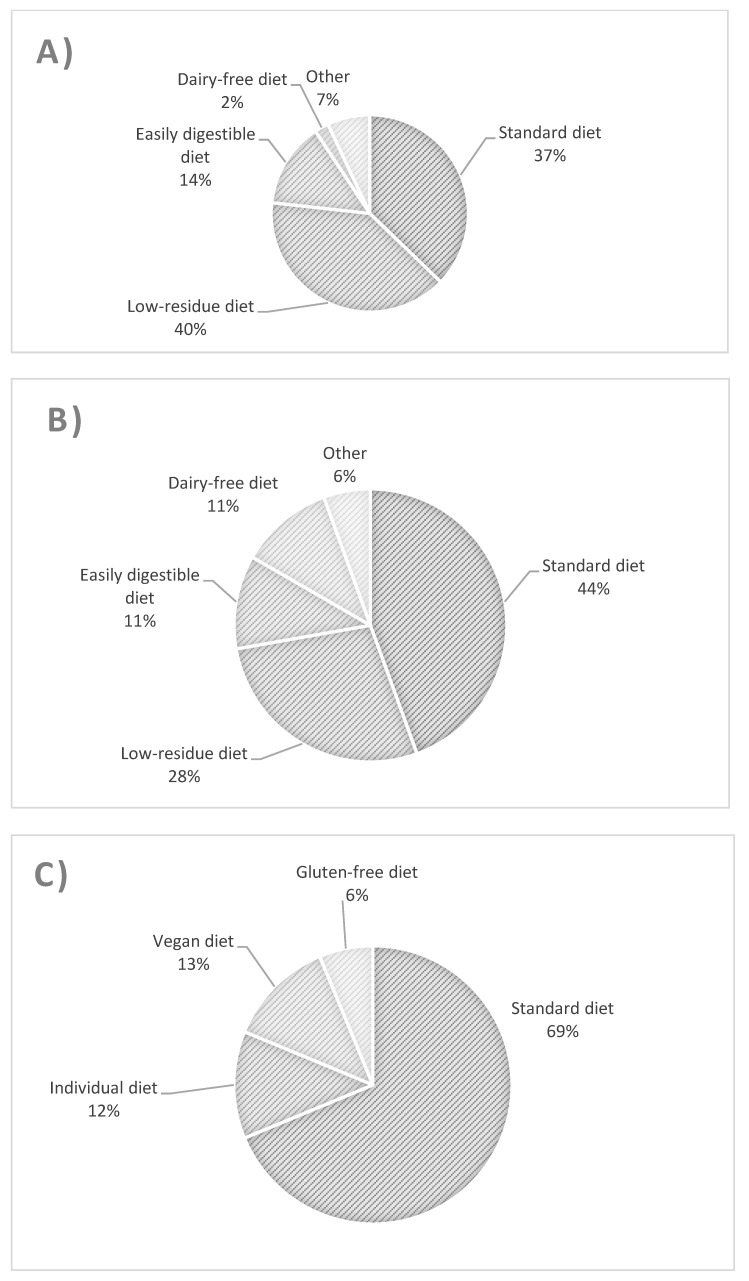
Types and percentage of diets used in patients with ulcerative colitis (**A**), Crohn’s disease (**B**), and controls (**C**).

**Figure 2 biology-11-00108-f002:**
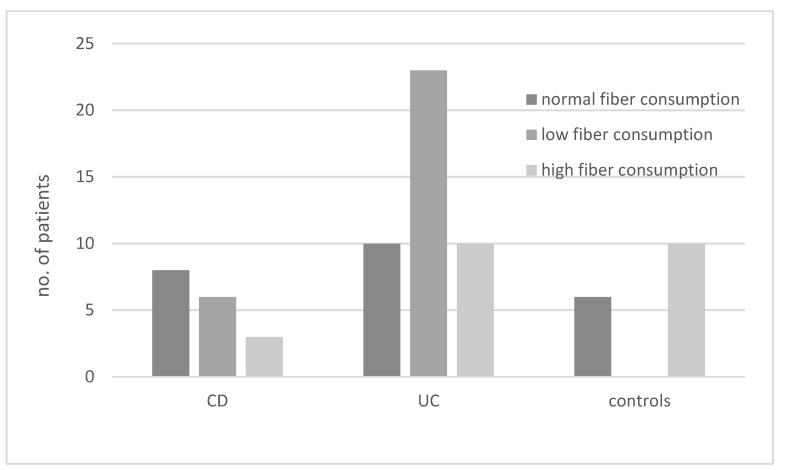
Consumption of diets with different fiber content (low, high, or normal) in patients with ulcerative colitis (UC), Crohn’s disease (CD), and controls.

**Figure 3 biology-11-00108-f003:**
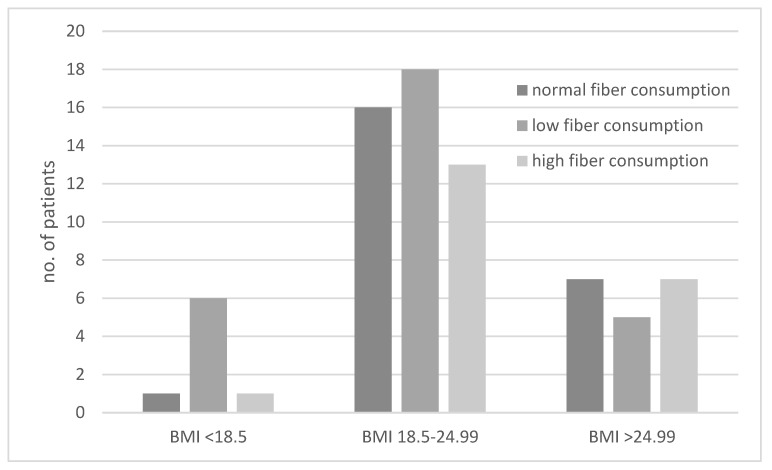
Comparison of dietary fiber intake in underweight (body mass index (BMI) < 18.5 kg/m^2^), normal-weight (BMI 18.5–24.99 kg/m^2^), and excessive-weight (BMI > 24.99 kg/m^2^) subjects.

**Figure 4 biology-11-00108-f004:**
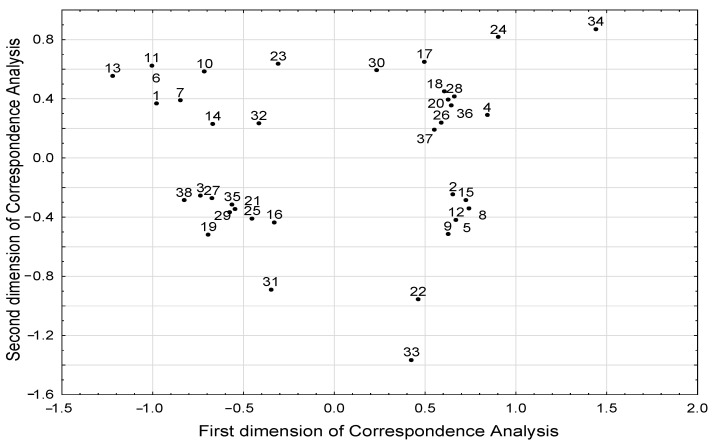
Structure of correlations between all parameters included in the correspondence analysis the projection of parameters in the space defined by first two dimensions of the correspondence analysis). Meaning of numeric symbols on the plot (the presence of the following phenomena): 1—age under the median 30 years; 2—age above the median 30 years; 3—no steroid treatment; 4—steroid treatment; 5—no alcohol consumption; 6—alcohol consumption; 7—standard diet; 8—easily digestible diet; 9—low-fiber diet < 25 g/day; 10—high-fiber diet > 30 g/day; 11—intake of whole-grain products every day; 12—lack of whole-grain products in diet; 13—consumption of legumes several times a week; 14—consumption of legumes several times a month; 15—lack of legumes in diet; 16—phosphoric acid concentration under the median 837.6 µg/g; 17—phosphoric acid concentration above the median 837.6 µg/g; 18—valeric acid concentration under the median 5.52 µg/g; 19—valeric acid concentration above the median 5.52 µg/g; 20—isovaleric acid concentration under the median 46.3 µg/g; 21—isovaleric acid concentration above the median 46.3 µg/g; 22—treated with other medications; 23—not treated with other medications; 24—weight loss; 25—stable body weight; 26—propionic acid concentration under the median 381.2 µg/g; 27—propionic acid concentration above the median 381.2 µg/g; 28—butyric acid concentration under the median 215.0 µg/g; 29—butyric acid concentration above the median 215.0 µg/g; 30—isobutyric acid concentration under the median 43.0 µg/g; 31—isobutyric acid concentration above the median 43.0 µg/g; 32—regular body weight; 33—overweight/obesity; 34—underweight; 35—no antibiotic treatment; 36—antibiotic treatment; 37—acetic acid concentration under the median 1135.3 µg/g; 38—acetic acid concentration above the median 1135.3 µg/g.

**Table 1 biology-11-00108-t001:** Characteristics of patients with ulcerative colitis (UC), Crohn’s disease (CD), and controls.

Parameter	UC	CD	Controls
no. of patients	43	18	16
sex (male/female)	29/14	10/8	3/13
age (min–max (median), years)	18–71 (32)	18–43 (30)	19–77 (23.5)
BMI (min–max, (median), kg/m^2^)	15.7–37.1 (21.6)	13.8–29.2 (20.2)	17–32.9 (21.6)

**Table 2 biology-11-00108-t002:** Association weights for FA levels and BMI categories, based on the correspondence analysis model (body mass index (BMI) overweight/obese—BMI > 25 kg/m^2^; BMI underweight—BMI < 18.5 kg/m^2^).

Pairs of Related Parameters		Correlation Weights
isobutyric acid > 43.0 µg/g	BMI overweight/obese	1.21
butyric acid < 215.0 µg/g	BMI underweight	0.95
isovaleric acid < 46.3 µg/g	BMI underweight	0.90
propionic acid < 381.2 µg/g	BMI underweight	0.84

The threshold values of fatty acid concentrations represent the median for a given parameter.

**Table 3 biology-11-00108-t003:** Association weights for age or fiber diet categories and consumptions of various plant products, regular diet, or BMI category, based on the correspondence analysis model (BMI underweight—BMI < 18.5 kg/m^2^).

Pairs of Related Parameters		Correlation Weights
age < 30 years	Consumption of legumes several times a week	1.20
age < 30 years	Consumption of whole-grain products everyday	0.97
high-fiber diet (>30 g/day)	Consumption of legumes several times a week	0.84
age < 30 years	Complete, regular diet	0.83
age < 30 years	BMI underweight	−1.39

The threshold values represent the median for a given parameter.

**Table 4 biology-11-00108-t004:** Association weights for BMI category or steroid treatment and consumptions of various plant products or antibiotics treatment, based on the correspondence analysis model (BMI underweight—BMI < 18.5 kg/m^2^).

Pairs of Related Parameters		Correlation Weights
BMI underweight	Steroid treatment	1.19
BMI underweight	Antibiotics treatment	0.92
steroid treatment	Intake of whole-grain products everyday	−0.82
steroid treatment	Consumption of legumes several times a week	−1.02

## Data Availability

The data presented in this study are available on request from the corresponding author.

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
