# Peer review of "Association between Fecal Short-Chain Fatty Acid Levels, Diet, and Body Mass Index in Patients with Inflammatory Bowel Disease"

_biology, 2022, doi:10.3390/biology11010108_

Round 1

Reviewer 1 Report

I have no further comment.

Author Response

Thank you for your review.

Reviewer 2 Report

This paper is well written and organized.  Please improve quality of all art work and  tabels to make them more interesting to readers.  Thank you

Author Response

Thank you for your valuable suggestions.

List of changes:

  1. The description of the axis of the coordinate system in Figure 4 has been modified (current version: “First dimension of Correspondence Analysis; Second dimension of Correspondence Analysis)”
  2. A caption has been added to the frame with explanations of symbols from Figure 4. (current version: “Meaning of numeric symbols on the plot (the presence of the following phenomena):”
  3. In Tables 2-4, the word "correlation" has been replaced with the word "association" for the sake of consistency in the text.
  4. The meaning of symbol CA was explained in section 2.4. Statistical Analysis.
  5. The caption to Figure 1. was slightly changed (current version: “Types and percentage of diets…”).

Reviewer 3 Report

About the work entitled: Association between Fecal Short-Chain Fatty Acid Levels, Diet, and Body Mass Index in Patients with Inflammatory Bowel Disease, I follow with some considerations:
-  The first statement (lines 44 to 46) that defines inflammatory bowel disease and correlates it with Chron and colitis, these diseases correspond to a portion of the inflammatory diseases that affect the intestine, they occur with prevalence, but they are not the only ones - reference is missing;
- In the introduction, insert information about the changes that occur with the intestinal microbiota (dysbiosis) in IBD;
- Generally speaking, the introduction needs to be better referenced, for example, between lines 72 and 82 there is only one citation - it is not appropriate to use a single citation to support an entire paragraph;
- In the introduction, bring epidemiological data on the increase in body weight index and its relationship with IBD, and with the increase in other diseases, for example. BMI only appears briefly in the purpose of the introduction;
- Although there are validated questionnaires, I ask the authors to indicate the validation of the application of self-reported questionnaires, as it needs the participants' commitment to respond faithfully and as this could compromise the results found (the last paragraph of the discussion is extremely timely and well-directed);
- How can the number of participants be considered a valid sample for the study conclusions?
- Figure and table captions should be more self-explanatory;
- About the time each patient was on the reported diet, was this considered? Was it investigated whether changes occurred after diagnosis and whether overweight was prior to the onset of the first sign/symptom of the disease? (make these issues more evident);
- Just a matter of fluidity of the text, remove repeated terms, such as: In our study - lines 356 and 362 (consider revising the language);
- Conclusion must be more incisive;
- Discussion should be more punctual, it is very long;

Author Response

Reviewer 3

About the work entitled: Association between Fecal Short-Chain Fatty Acid Levels, Diet, and Body Mass Index in Patients with Inflammatory Bowel Disease, I follow with some considerations:

  1. The first statement (lines 44 to 46) that defines inflammatory bowel disease and correlates it with Chron and colitis, these diseases correspond to a portion of the inflammatory diseases that affect the intestine, they occur with prevalence, but they are not the only ones - reference is missing;

Answer:

Thank you for your comment. As suggested, this sentence has been modified. We supplemented the manuscript with appropriate references.

  1. In the introduction, insert information about the changes that occur with the intestinal microbiota (dysbiosis) in IBD;

Answer:

Thank you for your suggestions. As suggested, we included additional information about dysbiosis in IBD in section 1. Introduction.

  1. Generally speaking, the introduction needs to be better referenced, for example, between lines 72 and 82 there is only one citation - it is not appropriate to use a single citation to support an entire paragraph;

Answer:

Thank you for your constructive comment. In section 1 Introduction, we inserted additional references.

  1. In the introduction, bring epidemiological data on the increase in body weight index and its relationship with IBD, and with the increase in other diseases, for example. BMI only appears briefly in the purpose of the introduction;

Answer:

Thank you for your suggestions. We have included more details on the issue of increasing the body mass index in patients with IBD. We have made the appropriate changes to the introduction.

  1. Although there are validated questionnaires, I ask the authors to indicate the validation of the application of self-reported questionnaires, as it needs the participants' commitment to respond faithfully and as this could compromise the results found (the last paragraph of the discussion is extremely timely and well-directed);

Answer:

In our study we used modified Food frequency questionnaire (FFQ), which is a dietary tool widely used in studies of nutritional epidemiology. It has many advantages such as cheapness, ease of filling, and providing important information on your intake over an extended period of time. Before completing the questionnaire, all patients were interviewed by a dietitian and instructed on how to complete the questionnaire in order to avoid mistakes. We are aware of the limitations associated with conducting questionnaire research. However, we are currently conducting research using 24 h interview as a nutritional analysis.

  1. How can the number of participants be considered a valid sample for the study conclusions?

Answer:

Thank you for your comment. Calculating the appropriate sample size is critical in medical studies. Therefore, we made a priori sample size estimates based on our preliminary research on IBD patients.

  1. Figure and table captions should be more self-explanatory;

Answer:

Thank you for your valuable suggestions. We improved figures and tables as suggested.

List of changes:

  1. The description of the axis of the coordinate system in Figure 4 has been modified (current version: “First dimension of Correspondence Analysis; Second dimension of Correspondence Analysis)”
  2. A caption has been added to the frame with explanations of symbols from Figure 4. (current version: “Meaning of numeric symbols on the plot (the presence of the following phenomena):”
  3. In Tables 2-4, the word "correlation" has been replaced with the word "association" for the sake of consistency in the text.
  4. The meaning of symbol CA was explained in section 2.4. Statistical Analysis.
  5. The caption to Figure 1. was slightly changed (current version: “Types and percentage of diets…”)

  1. About the time each patient was on the reported diet, was this considered? Was it investigated whether changes occurred after diagnosis and whether overweight was prior to the onset of the first sign/symptom of the disease? (make these issues more evident);

Answer:

Thank you for that comment. All subjects were asked about their current diet used for the last 3 months before the study. Patients who changed their eating habits at that time were excluded from the study. We also selected a group of patients with de novo diagnosed IBD, but we did not observe any correlations in this group (neither with diet nor with the concentration of organic acids). We also investigated the relationship between the symptoms (such as diarrhea, abdominal pain) and SCFA profile, but we found no associations.

  1. Just a matter of fluidity of the text, remove repeated terms, such as: In our study - lines 356 and 362 (consider revising the language);

Answer:

Thank you for your suggestion. We have made appropriate changes.

  1. Conclusion must be more incisive;

Answer:

Thank you for your suggestion. We have modified section 5. Conclusions.

  1. Discussion should be more punctual, it is very long;

Answer:

Thank you for your suggestion. We have made appropriate changes in manuscript.

Round 2

Reviewer 3 Report

The authors satisfactorily made the suggested changes